# Oxygen Tension Regulates Lysosomal Activation and Receptor Tyrosine Kinase Degradation

**DOI:** 10.3390/cancers11111653

**Published:** 2019-10-25

**Authors:** Jaewoo Hong, Todd R. Wuest, Yongfen Min, P. Charles Lin

**Affiliations:** Cancer and Inflammation Program, Center for Cancer Research, National Cancer Institute, Frederick, MD 21702, USA; toddwuest1@gmail.com (T.R.W.); yongfen.min@nih.gov (Y.M.)

**Keywords:** hypoxia, lysosome, oxygen tension, receptor tyrosine kinase, EGFR

## Abstract

Oxygen sensing is crucial for adaptation to variable habitats and physiological conditions. Low oxygen tension, or hypoxia, is a common feature of solid tumors, and hypoxic tumors are often more aggressive and resistant to therapy. Here we show that, in cultured mammalian cells, hypoxia suppressed lysosomal acidification/activation and receptor tyrosine kinase (RTK) degradation. Hypoxia down-regulated mTORc1, reducing its ability to activate transcription factor EB (TFEB), a master regulator of V-ATPase, the lysosomal proton pump. Hypoxia prevented epidermal growth factor receptor (EGFR) degradation in tumor tissues, whereas activation of lysosomes enhanced tumor cell response to anti-EGFR treatment. Our results link oxygen tension and lysosomal activity, provide a molecular explanation of the malignant phenotype associated with hypoxic tumors, and suggest activation of lysosomes may provide therapeutic benefit in RTK-targeted cancer therapy.

## 1. Introduction

Hypoxia is a condition in which the body, or a region of the body, is deprived of an adequate oxygen supply. It is a common stress associated with various pathological conditions, ranging from cardiovascular disorders to cancer, and affects many cellular and molecular activities, as well as a range of therapeutic responses [1,2,3]. Lysosomes are membrane-enclosed organelles containing abundant acid hydrolases. Lysosomes maintain an acidic environment principally by pumping protons in from the cytosol via the vacuolar H^+^-ATPase [4]. The lysosome is a major component of the cellular degradation machinery, digesting damaged and abnormal proteins, as well as directing the proteolysis of cellular enzymes and regulatory proteins that are no longer needed, in order to maintain cellular homeostasis [5,6]. Receptor tyrosine kinases (RTKs) are high affinity cell surface receptors for growth factors, cytokines, and hormones, playing roles in cell proliferation, differentiation and survival. Upon activation by ligand binding, receptors undergo endocytosis and lysosome-mediated receptor degradation to avoid excessive and prolonged signal activation [7,8].

## 2. Results

In studying the effect of hypoxia on RTK activity, we identified a link between hypoxia and receptor signaling. Human endothelial cells were cultured either in normoxic (20% O_2_) or hypoxic (1% O_2_) conditions in the presence or absence of growth factors. Ligand stimulation with recombinant epidermal growth factor (EGF) resulted in a reduction of EGF receptor (EGFR) levels in cells cultured in normoxia. Ligand stimulation failed to induce EGFR degradation in hypoxia (Figure 1A). We observed similar results in two epithelial tumor cell lines, HeLa and A549 (Figure 1A). Elevated EGFR levels in hypoxic conditions resulted in increased receptor phosphorylation and activation of downstream signaling in both types of cells (Appendix A). Hypoxia also blocked VEGF-induced VEGFR2 reduction in endothelial cells (Appendix A). These findings reveal a general inhibitory role of hypoxia in RTK degradation.

The failure of receptor degradation under hypoxic conditions suggested that hypoxia may play a role in lysosomal function. We used Bafilomycin A1, a specific inhibitor of V-ATPase, to block lysosomal acidification/activation under normoxic conditions and observed that EGF-induced EGFR degradation was reduced in cultured endothelial cells, as well as epithelial tumor cell lines after ligand stimulation, as seen in hypoxic conditions (Figure 1B). The addition of chloroquine (CQ), another lysosomal specific inhibitor, also blocked ligand-stimulation induced VEGFR2 reduction in endothelial cells (Appendix A). These results suggest that hypoxia may suppress receptor degradation through the inhibition of lysosomal activity.

We examined if hypoxia affects lysosome acidification. Using LysoBrite, a dye that is specific for acidic organelles, we observed a significant reduction in the LysoBrite signal in cells cultured in hypoxic conditions. The hypoxic treatment also led to a reduction of Lamp2, a marker for lysosomes (Figure 1C). The acidification of lysosomes is generated by the action of V-ATPase [9,10]. We measured two catalytic components of V-ATPase, Atp6v1a and Atp6v1b2, in cells incubated under hypoxic conditions and observed a gradual reduction of both subunits, which correlated with an increase of EGFR (Figure 1D). Thus, the inhibitory effect of hypoxia on lysosomal acidification is caused by the reduction of V-ATPase.

To further establish the connection between the acidification of lysosomes by V-ATPase with ligand-induced receptor degradation, we knocked down Atp6v1b2 expression in cells. Reduction of V-ATPase inhibited ligand-mediated receptor degradation, resulting in increased levels of EGFR (Figure 1E). Modest ectopic over-expression of Atp6v1a or Atp6v1b2 in endothelial cells did not result in a significant change in EGFR levels. Over-expression of both components led to a substantial reduction of EGFR (Figure 1F). In epithelial cells, ectopic expression of either one of these V-ATPase subunits led to a clear reduction of EGFR, and over-expression of both produced a more pronounced response than either one alone (Figure 1F). The difference observed between endothelial and epithelial cells may reflect the difference of endogenous V-ATPase subunit levels in different types of cells.

Hypoxia inhibits mTOR activity through the TSC1/2 complex and regulated in development and DNA damage response 1 (REDD1) [11,12,13], and the mTOR pathway is involved in controlling V-ATPase assembly during dendritic cell maturation [14]. We thus investigated if mTOR is responsible for hypoxia-mediated inhibition of V-ATPase expression. Hypoxia led to a reduction of both total mTOR and phosphorylated mTOR (Figure 2A). Blocking mTOR activity with rapamycin significantly reduced the levels of V-ATPase subunits, which correlated with increased EGFR levels (Figure 2B). To determine which mTOR complex regulates the expression of V-ATPase, we purified pulmonary microvascular endothelial cells from mTORc1 (Rheb^flox/flox^) or mTORc2 (Rictor^flox/flox^) floxed mice, and infected them with adenoviral vectors for the Cre recombinase. Deletion of Rheb (mTORc1) but not Rictor (mTORc2) resulted in a reduction of Atp6v1a and Atp6v1b2 (Figure 2C). Treating endothelial cells with the mTORc1-specific inhibitors rapamycin and AZD2014 led to reduction of the acidification of lysosomes (Figure 2D). Activation of mTORc1 using MHY1485 increased acidification of lysosomes (Figure 2E). These results suggest that hypoxia inhibits lysosomal acidification/activation through suppression of mTORc1 mediated V-ATPase expression.

V-ATPase regulation by mTORc1 involves TFEB [15], which binds to the regulatory regions of many lysosomal genes, activating their expression [16]. TFEB shuttles between the cytosol and nucleus, and nuclear translocation is required for its function. In normoxic cells, immunostaining for TFEB protein revealed the majority was in the nucleus, whereas it was mainly in the cytosol under hypoxic conditions, as was seen when cells were cultured in normoxia in the presence of an mTORc1 specific inhibitor (Appendix A). Addition of the mTORc1 activator, MHY1485, reversed the hypoxia-mediated blockade of TFEB nuclear translocation (Appendix A). These findings suggest that hypoxia inhibits TFEB nuclear translocation via inhibition of mTOR, thereby preventing V-ATPase transcription.

TFEB phosphorylation regulates nuclear translocation. Both Serine 142 (S142) [17], and a serine-rich region at the C-terminus (_462_SSRRSSFS_469_) [15] have been implicated in TFEB nuclear translocation. To determine which site(s) is(are) responsible for hypoxia-mediated translocation, we transfected cells with phospho-mimic or inhibitory mutants of TFEB, and subjected them to normoxia or hypoxia. Mutating S142 to either alanine or aspartic acid had no effect on TFEB nuclear translocation in response to hypoxia. Mutating all five serine residues at the C-terminus to alanines blocked TFEB nuclear translocation. Mutating them to aspartic acids led to spontaneous nuclear translocation of TEFB regardless the levels of oxygen (Figure 3A). To determine which serine residue(s) in this cluster is phosphorylated by mTORc1, we over-expressed Myc-tag TFEB in HeLa cells, followed by treatment with either vehicle, or MHY1485, or AZD2014. TFEB protein was pulled down with an anti-Myc antibody and subjected to mass-spectrometry. Phosphorylation at S462 was detected in all three treatment groups, while phosphorylation at S463 only occurred after a stimulation with mTORc1 activator MHY1485 (Figure 3B). There was no detectable phosphorylation signal on any other serine residue in this cluster. S142 in TFEB was phosphorylated regardless the activation status of mTORc1 (Appendix A). Thus, mTORc1 specifically phosphorylates TFEB at the Serine 463 site.

To evaluate the role of phosphorylation at S463 on TFEB nuclear translocation, we mutated this serine to either alanine (S463A) or aspartic acid (S463D). As a control, we also mutated S462 to aspartic acid (S462D). The mutant constructs were transfected into HeLa cells followed by treatment with AZD2014. Blocking mTORc1 inhibited the nuclear translocation of wild-type TFEB, as well as the S462D mutant. However, AZD2014 failed to block S463D nuclear translocation (Figure 3C). Transfection of the S463A and S463D mutants into HeLa cells, and their exposure to either normoxic or hypoxic conditions, revealed the S463A mutation blocked TFEB nuclear translocation, and the S463D mutant spontaneously translocated to the nucleus, independent of oxygen tension (Figure 3D). Activation of mTORc1 with MHY1485 reversed the hypoxia-induced nuclear blockade of wild-type TFEB. The S463D mutant spontaneously translocated to the nucleus and the S463A mutant failed to translocate to the nucleus regardless of oxygen tension and mTOR activity (Appendix A). Thus, mTORc1 phosphorylates TFEB at S463, and this phosphorylation is necessary and sufficient for hypoxia-mediated TFEB nuclear translocation.

Hypoxia is a hallmark of solid tumors and is positively correlated with invasive phenotypes, resistance to therapy and poor prognosis of cancer patients [2,18,19,20]. RTKs are major drivers in cancer growth and progression, and RTK inhibitors are commonly used in cancer therapy. Since hypoxia blocks RTK degradation, hypoxic tumors may have elevated RTK levels that contribute to the phenotypes associated with poor prognosis. Analyzing murine B16 melanoma tumor tissues, we found a positive correlation between hypoxia, measured by HIF-1α accumulation, and EGFR levels. The hypoxic region had significantly higher levels of EGFR than the less hypoxic region of the tumor (Figure 4A). Analysis of human cancer tissues also revealed a positive correlation between EGFR levels and hypoxia in both colon and stomach cancer samples (Figure 4B). In addition, there is a positive correlation of hypoxia with phosphorylation of ERK, a downstream mediator of EGFR signaling (Figure 4C). To determine what drives elevated EGFR levels in hypoxic tumors, we incubated three different tumor cell lines plus one endothelial cell line in hypoxia or normoxia and measured transcription by RT-qPCR. There was no statistical increase in EGFR transcript levels in the four cell lines, regardless of the hypoxic or nontoxic conditions. *VEGF*, a gene known to be regulated by hypoxia, was increased in all four cell lines subjected to hypoxia (Figure 4D). These findings suggest that increased RTK levels in hypoxic tumors are likely due to inhibited receptor degradation in lysosomes.

EGF signaling is involved in cell proliferation and cancer progression, and EGF inhibitors are widely used in cancer therapy. Given that hypoxia blocked lysosome-mediated EGFR degradation, resulting in increased signaling, lysosomal activators should increase receptor degradation and potentially provide therapeutic benefit in anti-RTK cancer therapy. To evaluate this idea, we ectopically expressed the wild-type and S463D mutant of TFEB in three human cancer cell lines to modulate lysosomal acidification/activation and treated the cells with EGFR inhibitors under hypoxia. S463A mutant was included as a control. Activation of lysosomes with the S463D significantly reduced cell proliferation in all three tumor lines. A combination of lysosomal activation and EGFR inhibitors produced the strongest inhibition of all tumor cells (Figure 4E). In contrast, there is no significant difference between cells transfected with the WT or the S463A mutant, consistent with the finding that hypoxia blocks TFEB phosphorylation and nuclear translocation.

## 3. Discussion

We demonstrate a critical function for oxygen tension in regulating lysosomal activation and proteolysis via the mTORc1–TFEB pathway. Hypoxia occurs during normal mammalian development and is involved in developmental morphogenesis. It is also associated with various pathological disorders, including ischemic cardiovascular diseases, stroke, and cancer. Lysosomes are the terminal organelles on the endocytic pathway, serving both to degrade material taken up from outside the cell, as well as biological polymers inside cells. We reveal an inhibitory function for hypoxia in lysosomal acidification, required for the activation of various hydrolytic enzymes responsible for breaking down biological polymers. Thus, hypoxia blocks lysosomal activation and function, and this has broad implications for cell regulation and homeostasis under hypoxic stress. RTKs are major drivers in cancer development and progression. Persistent activation of EGFR enables cancer cells to engage in autonomous proliferation, which is a critical hallmark of cancer [21]. EGFR expression is a marker of advanced tumor stages, resistance to standard therapeutic approaches, and reduced patient survival [22]. We show that hypoxic conditions suppress EGFR degradation in lysosomes and results in elevated signaling. Levels of EGFR and its downstream signaling are positively correlated with levels of hypoxia in both murine and human tumor tissues, suggesting that hypoxic tumors experience prolonged and enhanced signaling, allowing the tumor cells to survive and maintain homeostasis under stress. These findings may explain why hypoxic tumors tend to be more malignant, associated with poor prognosis, and are often resistant to therapy [19,20,23]. Our data suggest that RTK inhibitors may deliver a more potent therapeutic effect when combined with lysosomal activators. Initial analysis demonstrated that increase of lysosomal acidification could enhance or sensitize tumor cell response to an EGFR inhibitor, potentially providing a strategy for targeted cancer therapy.

## 4. Materials and Methods

### 4.1. Experimental Animals

The mice were maintained in pathogen-free facilities in the National Cancer Institute (Frederick, MD, USA). The study was approved by the NCI Animal Care and Use Committee (protocol number 17-009, 17-010 and 17-048), and in accordance with the Animal Research: Reporting of In Vivo Experiments (ARRIVE) guidelines. Rheb- and Rictor-flox mice are on a C57BL/6 background. Age and sex matched mice were used in isolation of endothelial cells, and pooled cells from 3–5 mice per group were used. 5 × 10^5^ of B16 cells were injected to the right flank of C57BL/6 mice. The tumor tissue was harvested between 3–5 weeks post-injection.

### 4.2. Cell Culture and Bioassays

Human umbilical vein endothelial cells (HUVECs) and human epithelial cell lines (HeLa, A549, and DLD-1) were obtained from Lonza (Walkersville, MD, USA) and ATCC (Manassas, VA, USA), respectively. Lung endothelial cells were isolated from Rheb- and Rictor-floxed mice as described [24]. The cells were cultured according to the manufacture’s protocols. Recombinant EGF and VEGF proteins were purchased from ProSpec (East Brunswick, NJ, USA). For EGFR and VEGFR2 degradation assays, serum starved cells were stimulated with 50 ng/mL of EGF or 100 ng/mL of VEGF for 5 h. For hypoxia, cells were incubated in 1% O_2_ for 1–6 h. mTORc1 inhibitor, Rapamycin (Selleckchem, Houston, TX, USA) or AZD2014 (Selleckchem) at 10 nM, and mTORc1 activator, MHY1485 at 2 μM, were used. Cell transfection of Myc tagged TFEB and V-ATPase components, pcDNA-ATP6V1A plus pcDNA-ATP6V1B2 were carried out by using Fugene HD (Promega, Madison, WI, USA). For the anti-EGFR assay, cells were treated with 10 μg/mL of anti-EGFR monoclonal antibody (Sigma, clone LA1, St. Louise, MO, USA) or control mouse IgG for 16 h. XTT cell proliferation assay was conducted following the description.

### 4.3. Immunofluorescent Staining and Microscopy

Cells were incubated with LysoBrite-NIR (AAT Bio, Sunyvale, CA, USA) following the manufacturers protocol. Cells were fixed in 4% paraformaldehyde, permeabilized with 0.1% Triton X-100, and incubated with a primary antibody followed by fluorescent dye-conjugated secondary antibody. Antibodies against Lamp2 and Myc tag for TFEB were obtained from DSHB and Cell Signaling, respectively. Alexa 488-conjugated wheat germ agglutinin (Thermo Fisher Scientific, Waltham, MA, USA) was used to stain the membranous structure. The tumor tissue was stained with antibodies against EGFR (Sigma-Aldrich, St. Louise, MO, USA), HIF-1α (Abcam, Cambridge, UK) and p-ERK (Cell Signaling, Danvers, MA, USA). Human tumor tissue slides were purchased from Genetex (Irvine, CA, USA).

Confocal microscopy images were analyzed with Zen software (Carl Zeiss, Oberkochen, Germany) and imageJ (NIH, Bethesda, MD, USA).

### 4.4. Western Blot and RT-qPCR

Cells were lysed with sample buffer and briefly ultrasonicated. SDS-PAGE and Western blot was carried out on 5 μg of total protein. Specific antibodies for EGFR (#2085), VEGFR2 (#9698), p-mTOR (#5536), mTOR (#2983), p-S6K (#9204), p-Akt (#4060), Rheb (#13879), Rictor (#9476) were purchased from Cell Signaling Technology (Danvers, MA) and antibodies against Atp6v1a (#GT3846) and Apt6v1b2 (#GTX110783) from GeneTex (Irvine, CA, USA). Isolated by RNeasy mini (Qiagen, Hilden, Germany), 100 μg of total RNA was propagated to RT-qPCR from cell lines. Relative mRNA level was measured with primers purchased against EGFR (#KSPQ12012, Sigma-Aldrich) and VEGF (#KSPQ12012, Sigma-Aldrich).

### 4.5. Statistics

Prism (Graphpad) was used for all statistical analyses. For analysis, two-way ANOVA was used followed by Bonferroni Multiple Comparisons test.

## 5. Conclusions

This study demonstrates that hypoxia suppresses lysosomal activity and results in elevated RTK levels and signaling. These findings may explain why hypoxic tumors tend to be more malignant and associated with poor prognosis. It potentially provides a new strategy for targeted cancer therapy when combining RTK inhibitors with lysosomal activators.

## Figures and Tables

**Figure 1 cancers-11-01653-f001:**
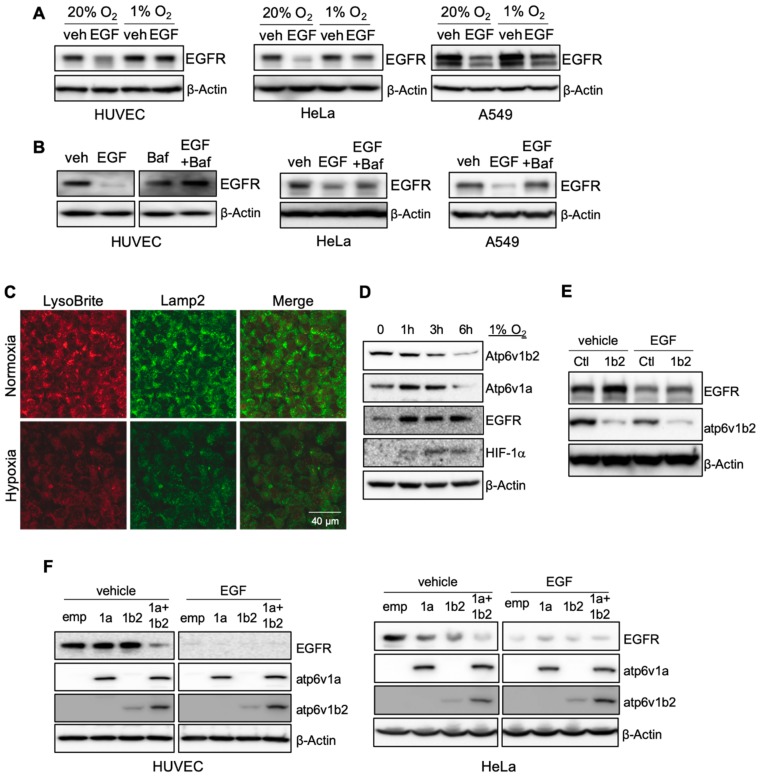
Hypoxia inhibits lysosomal activity via suppression of V-ATPase expression. HUVECs, HeLa and A549 were serum-starved overnight, followed by stimulation with 50 ng/mL of epidermal growth factor (EGF) under 20% or 1% oxygen conditions for 5 h. Cell lysates were used to measure epidermal growth factor receptor (EGFR) levels by Western blot (**A**). Serum starved cells were treated with 50 ng/mL of EGF in the presence or absence of 100 nM of Bafilomycin A1 for 5 h, followed by Western blot for EGFR (**B**). HUVECs were incubated in either 20% or 1% O_2_ in the presence of LysoBrite-NIR for 4 h. Cells were fixed and immunostained for Lamp2 (**C**). Images were collected from confocal microscopy. Representative images are shown. HUVECs were cultured in 1% O_2_ for the indicated times, followed by Western blot (**D**). HUVECs were infected with ATP6V1B2 shRNA lentiviral or control vectors (**E**), or infected with lentiviral vectors for ATP6V1A and/or ATP6V1B2, for 2 days (**F**). The cells were serum-starved overnight followed by stimulation with 50 ng/mL of EGF for 5 h. Appendix A shows the raw data for Western blots of Figure 1.

**Figure 2 cancers-11-01653-f002:**
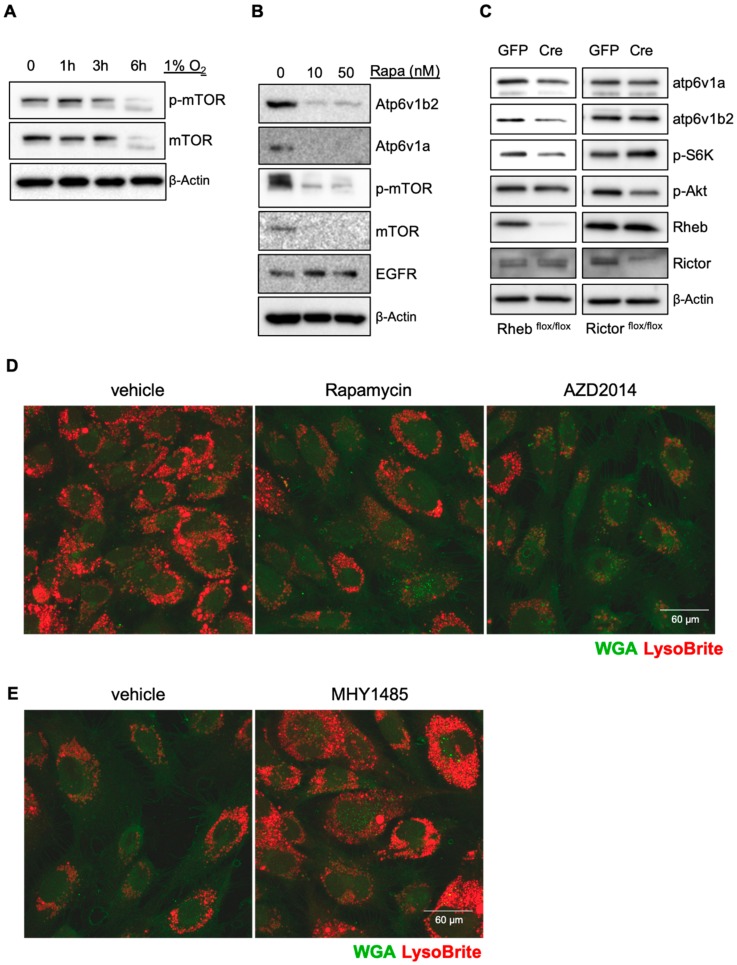
Hypoxia suppresses lysosomal acidification through mTORc1. HUVECs were incubated in 1% O_2_ for the indicated times, and phospho- and total mTOR levels were detected from cell lysates (**A**). HUVECs were treated with 10 or 50 nM of rapamycin overnight. Cell lysates were subjected to Western blot (**B**). Lung endothelial cells were isolated from *RHEB*- or *RICTOR*-floxed mice followed by infection with GFP or Cre expressing adenoviral vectors for 24 h, followed by Western blot (**C**). HUVECs were incubated in the presence of LysoBrite-NIR in the presence or absence of 10 nM of rapamycin or AZD2014 to block mTORc1(**D**), or in the presence or absence of 2 μM of MHY1485 to activate mTORc1(**E**), overnight. Wheat germ agglutinin (WGA) was used for counter staining. Images were collected from confocal microscopy. Representative images are shown. Appendix A shows the raw data for Western blots of Figure 2.

**Figure 3 cancers-11-01653-f003:**
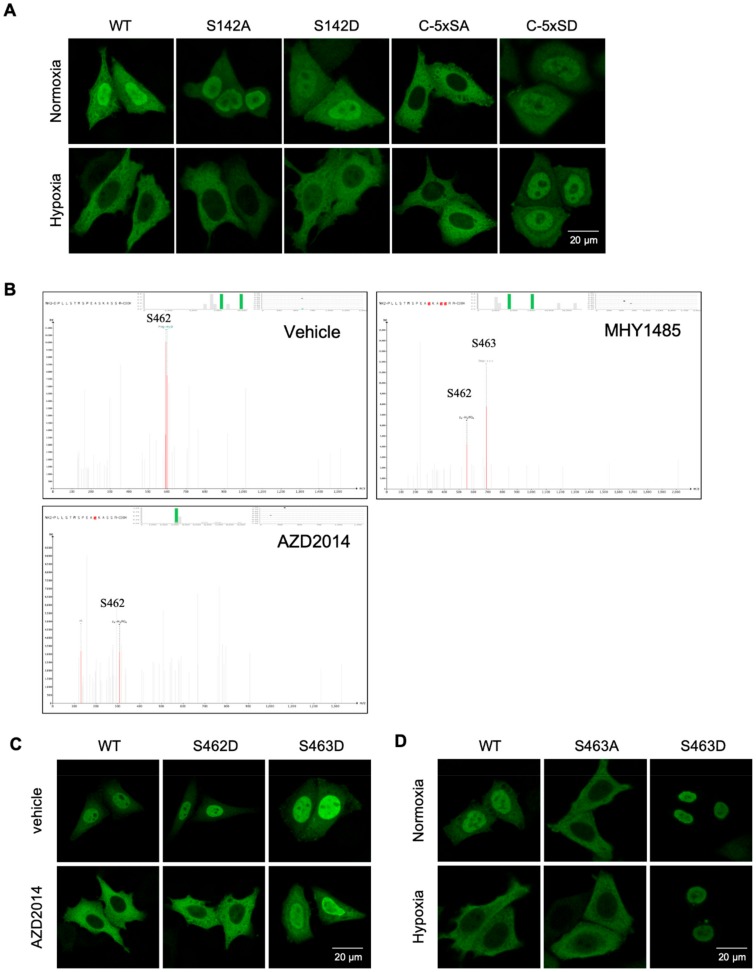
mTORc1 phosphorylates TFEB at S463. HeLa cells transfected with Myc tagged wild-type (WT) or TFEB mutant constructs were incubated in either 20% or 1% O_2_ for 24 h, followed by immunostaining using Myc antibody (**A**). HeLa cells transfected with TFEB-Myc expression vectors for 1 day, followed by stimulation with vehicle control, 2 μM of MHY1485, or 10 nM of AZD2014 for 30 min. TFEB was immunoprecipitated with antibodies against the Myc tag, followed by mass-spectrometry analysis (**B**). HeLa cells transfected with S462D or S463D mutants were stimulated with vehicle or 10 nM of AZD2014 for 30 min, followed by immunostaining for TFEB (**C**). HeLa cells transfected with S463A or S463D mutants were incubated in 20% or 1% O_2_ for 4 h, followed by immunostaining for TFEB (**D**). Images were collected from confocal microscopy. Representative images are shown.

**Figure 4 cancers-11-01653-f004:**
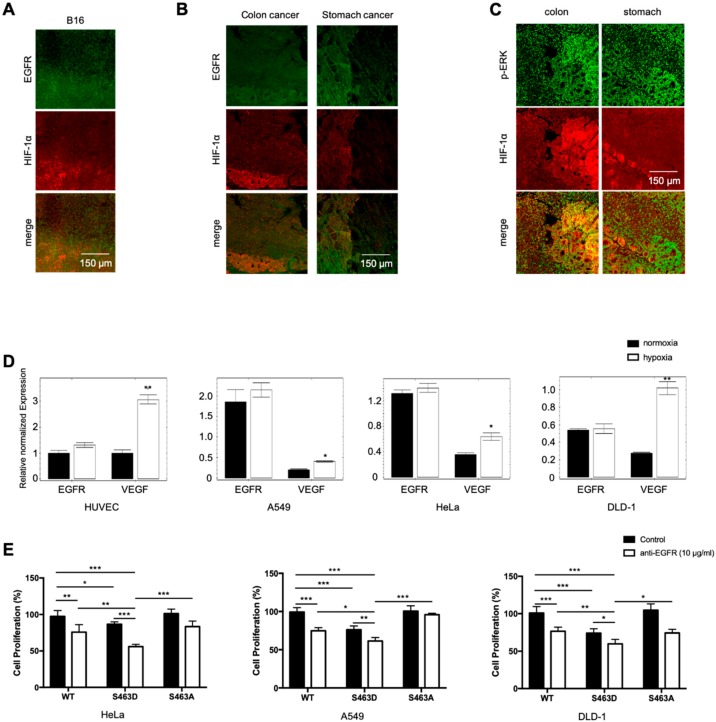
Activation of lysosomes enhances tumor cell response to anti-EGFR inhibition. Cryosectioned murine B16 melanoma tissues were stained with antibodies against EGFR and HIF-1α and imaged with confocal microscopy (**A**). Paraffin sections of human colon and stomach cancer tissues were stained against EGFR and HIF-1α (**B**) and p-ERK and HIF-1α (**C**) and imaged with confocal microscopy. Representative images are shown. HUVEC, A549, HeLa, and DLD-1 were incubated in 20% or 1% O_2_ for 5 h. The mRNA levels of EGFR and VEGF were measured by qPCR (**D**). * *p* < 0.05, ** *p* < 0.01. Human tumor cells transfected with WT TFEB, or S463A and S463D mutant expression vectors were treated with control IgG or anti-EGFR antibodies at 10 μg/mL overnight. Cell proliferation was measured by the XTT assay (**E**). * *p* < 0.05, ** *p* < 0.01, *** *p* < 0.001. The experiment was done in triplicates and repeated twice.

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
