# Peer review of "Oxygen Tension Regulates Lysosomal Activation and Receptor Tyrosine Kinase Degradation"

_cancers, 2019, doi:10.3390/cancers11111653_

Round 1
Reviewer 1 Report
I liked this paper. It seems thorough and merits publication. It would be nice if the authors were willing to consider adding a few references which place their work in the wider field of hypoxia-induced drug resistance. Examples might be:
Ke Xu et. al., Molecular Therapy 27(10), pages 1810-, (2019) Jae-Young Kim et. al., Int. J. Mol. Sci. 2017, 18, 1854-. Qinglong Guo et. al., Cell Death and Disease (2018) 8, e3178- Barbara Muz et. al., Hypoxia 2015(3), 83-92, and others
Author Response
Hi Reviewer,
I really appreciate your positive comments.
I have added some corrections in the introduction to add a bit of disease relevance to hypoxia.
I added a short statement regarding the hypoxic tumor, which is the biggest outlook for us.
Once more, I appreciate your review.
Thanks,
Jaewoo
Reviewer 2 Report
In this study, Jaewoo Hong et al., have provide the evidence that hypoxic condition is able to modulate lysosomal activation and plays inhibitory role on RTK degradation, providing the evidences that therapeutic efficiency could be enhanced if combined RTK inhibitors with lysosomal activators.
The rationale of this study as well as the experimental design is well presented. The results good support the hypothesis. However, some minor revisions are needed to improve paper quality before accept it for publication.
I would suggest to deepen more the introduction section, regarding the general involvement of hypoxia in some different tumors as well the current therapy might be taken into account as well
Comment 1. In Materials and Methods section, 4.4 Western blot and qPCR, catalogue number of all reagents should be provided.
Comment 2. In Materials and Methods section, 4.4 Western blot and qPCR, the authors should be indicate the protein quantity used to perform electrophoresis.
Comment 3. The quality of western blots is good, however all immunoblots showed in Figures 1, 2 and in supplementary data should be support by densitometric analysis and subsequent statistical assessment to certify the correct interpretation.
Author Response
Hi Reviewer,
I really appreciate your positive comments on my manuscript.
Your suggestions would make my manuscript much greater and perfect.
Below you will find the point-to-point answers on your comments.
---
I would suggest to deepen more the introduction section, regarding the general involvement of hypoxia in some different tumors as well the current therapy might be taken into account as well
=>I added some more background about the hypoxic tumor and disease relavance. Since this manuscript has been written as a report form, I did not put too much information in the introduction.
Comment 1. In Materials and Methods section, 4.4 Western blot and qPCR, catalogue number of all reagents should be provided.
=> Fixed
Comment 2. In Materials and Methods section, 4.4 Western blot and qPCR, the authors should be indicate the protein quantity used to perform electrophoresis.
=>Fixed
Comment 3. The quality of western blots is good, however all immunoblots showed in Figures 1, 2 and in supplementary data should be support by densitometric analysis and subsequent statistical assessment to certify the correct interpretation.
=>Fixed
Reviewer 3 Report
This is a well designed study on why hypoxia suppresses lysosomal activation and receptor tyrosine degradation. A variety of techniques are used in the study. The findings are interesting, the results are convincing, and the manuscript is well written.
Minor comment: P2. line 70-74, font size does not match the rest of the manuscript.
Author Response
Dear Reviewer,
I appreciate your positive comments.
Your concern about the font size difference has been double-checked during the proofreading process.
Thank you so much,
Jaewoo